# Improvement of Time Forecasting Models Using Machine Learning for Future Pandemic Applications Based on COVID-19 Data 2020–2022

**DOI:** 10.3390/diagnostics13061121

**Published:** 2023-03-15

**Authors:** Abdul Aziz K Abdul Hamid, Wan Imanul Aisyah Wan Mohamad Nawi, Muhamad Safiih Lola, Wan Azani Mustafa, Siti Madhihah Abdul Malik, Syerrina Zakaria, Elayaraja Aruchunan, Nurul Hila Zainuddin, R.U. Gobithaasan, Mohd Tajuddin Abdullah

**Affiliations:** 1Faculty of Ocean Engineering Technology and Informatics, Universiti Malaysia Terengganu, Kuala Nerus 21030, Terengganu, Malaysia; 2Special Interest Group on Applied Informatics and Intelligent Applications (AINIA), Universiti Malaysia Terengganu, Kuala Nerus 21030, Terengganu, Malaysia; 3Special Interest Group on Modeling and Data Analytics (SIGMDA), Universiti Malaysia Terengganu, Kuala Nerus 21030, Terengganu, Malaysia; 4Faculty of Electronic Engineering & Technology, Pauh Putra Campus, Universiti Malaysia Perlis (UniMAP), Arau 02600, Perlis, Malaysia; 5Centre of Excellence for Advanced Computing, Pauh Putra Campus, Universiti Malaysia Perlis (UniMAP), Arau 02600, Perlis, Malaysia; 6Faculty of Science, Institute of Mathematical Sciences, Universiti Malaya, Kuala Lumpur 50603, Malaysia; 7Mathematics Department, Faculty of Science and Mathematics, Universiti Pendidikan Sultan Idris, Tanjong Malim 53900, Perak Darul Ridzuan, Malaysia; 8Faculty of Fisheries and Food Science, Universiti Malaysia Terengganu, Kuala Nerus 21030, Terengganu, Malaysia; 9Fellow Academy of Sciences Malaysia, Level 20, West Wing Tingkat 20, Menara MATRADE, Jalan Sultan Haji Ahmad Shah, Kuala Lumpur 50480, Malaysia

**Keywords:** COVID-19 pandemic, machine learning, hybrid models, forecasting, public health, accuracy and efficiency

## Abstract

Improving forecasts, particularly the accuracy, efficiency, and precision of time-series forecasts, is becoming critical for authorities to predict, monitor, and prevent the spread of the Coronavirus disease. However, the results obtained from the predictive models are imprecise and inefficient because the dataset contains linear and non-linear patterns, respectively. Linear models such as autoregressive integrated moving average cannot be used effectively to predict complex time series, so nonlinear approaches are better suited for such a purpose. Therefore, to achieve a more accurate and efficient predictive value of COVID-19 that is closer to the true value of COVID-19, a hybrid approach was implemented. Therefore, the objectives of this study are twofold. The first objective is to propose intelligence-based prediction methods to achieve better prediction results called autoregressive integrated moving average–least-squares support vector machine. The second objective is to investigate the performance of these proposed models by comparing them with the autoregressive integrated moving average, support vector machine, least-squares support vector machine, and autoregressive integrated moving average–support vector machine. Our investigation is based on three COVID-19 real datasets, i.e., daily new cases data, daily new death cases data, and daily new recovered cases data. Then, statistical measures such as mean square error, root mean square error, mean absolute error, and mean absolute percentage error were performed to verify that the proposed models are better than the autoregressive integrated moving average, support vector machine model, least-squares support vector machine, and autoregressive integrated moving average–support vector machine. Empirical results using three recent datasets of known the Coronavirus Disease-19 cases in Malaysia show that the proposed model generates the smallest mean square error, root mean square error, mean absolute error, and mean absolute percentage error values for training and testing datasets compared to the autoregressive integrated moving average, support vector machine, least-squares support vector machine, and autoregressive integrated moving average–support vector machine models. This means that the predicted value of the proposed model is closer to the true value. These results demonstrate that the proposed model can generate estimates more accurately and efficiently. Compared to the autoregressive integrated moving average, support vector machine, least-squares support vector machine, and autoregressive integrated moving average–support vector machine models, our proposed models perform much better in terms of percent error reduction for both training and testing all datasets. Therefore, the proposed model is possibly the most efficient and effective way to improve prediction for future pandemic performance with a higher level of accuracy and efficiency.

## 1. Introduction

The city of Wuhan in Hubei Province, China, made history as the first point of the spread of the coronavirus disease (COVID-19) due to severe respiratory syndrome. On January 31, the World Health Organization (WHO) declared for the first time that COVID-19 is a “public health emergency of international concern” [1]. The virus was originally thought to have come from a fish market in Wuhan. On 11 January 2020, the gene sequence that China openly shared through personal contacts fueled its rapid spread, with a total of 9,129,146 confirmed cases, including 473,797 deaths worldwide as of 24 June 2020 [2]. However, as of 1 May 2021, COVID-19 has infected more than 151 million people and caused three million deaths worldwide. Countries such as the USA, Brazil, Russia, Spain, UK, Italy, France, Germany, China, India, Iran, and Pakistan have been hit the hardest by COVID-19. The first cases of COVID-19 reported in Malaysia on 2 January 2020 were detected by Chinese tourists entering the country from Singapore [3]. Only single-digit daily cases were reported in the initial phase, but this increased to 235 by 26 March [4]. The number of daily cases in Malaysia continued to increase exponentially by reaching around 20,000 in August 2021. The Malaysian government declared the implementation the Movement Control Order (MCO) from 18 March to 3 May 2020, the Conditional MCO (CMCO) from 4 May to 9 June 2020, and the Recovery MCO (RMCO) from 10 June 2020 to 31 March 2021. All travel and socio-economic activities (religious and cultural gatherings were not allowed) have been restricted across the country to keep new infections at bay and avoid overloading the country’s healthcare system during this time. All government and private offices and educational institutions, including transportation hubs, have been closed, citizens have been ordered to stay at home, and interstate travel has been banned, with fines of up to RM 10,000 for violators.

Since the WHO declared the COVID-19 outbreak a pandemic, not only governments from around the world, but also dedicated medical institutions have made many efforts to find vaccines and treatments to control the spread of the virus. Statisticians and public health scientists have also performed extensive statistical modelling, especially regarding the forecasting of COVID-19 cases, to help the health system prevent the contagion catastrophe. In this scenario, the ability to most effectively determine the growth rate at which the epidemic is spreading is very critical to counterattack and help governments, social planning, and policy making to accurately address the epidemic. Therefore, the motivation behind this research compared to the existing research is to (i) develop the most accurate and efficient predictive model related to the spread of COVID-19 in Malaysia and (ii) compare the performance of this new model with autoregressive integrated moving average (ARIMA), support vector machine (SVM), least-squares support vector machine (LSSVM), autoregressive integrated moving average-support vector machine ARIMA–SVM, and autoregressive integrated moving average–least-squares support vector machine models (ARIMA–LSSVM).

During the pandemic, many studies have been conducted using various mathematical and statistical models to predict the spread of the COVID-19 pandemic. One of the most popular time-series prognostic models for analyzing and predicting disease spread is the ARIMA (*p*, *d*, *q*) model [5,6,7]. Predicting new daily cases of COVID-19 was a difficult task, as cases increased daily. In the first wave, the pattern of COVID-19 cases has been continuously increasing for a period and then decreasing. However, for the second wave, it appears to be picking up again, and some of the COVID-19 cases are difficult to predict. In this scenario, some researchers predict the pattern of COVID-19 using ARIMA [8,9,10,11,12,13,14]. However, the ARIMA model has a limitation in that it typically can only handle a linear time-series data structure [15]. ARIMA model approximations are insufficient to pose a time-series prediction obstacle for researchers, especially for nonlinear patterns [16]. Despite its superior performance, the classification performance of Support Vector Machines (SVMs) and the generalizability of the classifier are often affected by the dimension or number of feature variables used, as mentioned by Lee [17]. As a result of the development of Vector Machines models, this process will be able to provide the most accurate and efficient result in each prediction case. SVMs, first introduced in 1995 by Vladimir Vapnik [18] in the field of statistical learning theory and structural risk minimization, have proven useful in a variety of prediction problems and classifications. SVMs could also manage or address difficulties such as non-linearity, local minimum, and high dimension where the ARIMA model could not [15,19,20,21]. SVM models have recently been used to handle problems such as nonlinear, local minimum, and high dimension. SVM can even guarantee higher accuracy for long-term predictions compared to other computational approaches in many practical applications. However, the single SVM model as a single ARIMA model also has some limitations, as the SVM model can only handle non-linear data and not linear data. With the limitations of a single ARIMA and SVM model, as well as an in-depth analysis of time-series prediction, hybrid approaches have become the best approach to overcome both limitations, and have a very significant impact in many areas due to their dynamic nature and higher level of predicting accuracy, efficiency, and precision. This approach is crucial because of the problems encountered in time-series forecasting, where almost all real time series contain linear and nonlinear correlation patterns between the data. Recently, the hybridization of prediction methods has been used with great success to achieve higher prediction accuracy [15,16,19,20,22,23,24,25,26].

Regarding the spread of COVID-19, the hybrid time-series model approach is crucial for predicting the impact of the COVID-19 outbreak, and has proven successful in predicting COVID-19 [27,28,29,30,31,32,33]. This study aims to (a) propose the ARIMA–LSSVM hybrid model approach to achieve better forecast results when it is able to produce the best estimator, i.e., produce small error terms; additionally, it aims to (b) examine the performance of the proposed models by comparing them to ARIMA and SVM models using three daily cases of COVID-19 data in Malaysia, that is, daily new positive cases, daily new deaths, and daily new recovered cases. Despite recent advances in time series and on COVID-19, the modelling process does not include COVID-19 cases specifically in Malaysia to help authorities manage the spread of this outbreak by producing more efficient, more accurate data, and more accurate forecasting results.

This study makes a significant contribution to the field of pandemic prediction and prevention by introducing novel approaches to dealing with COVID-19 data. Rather than relying on traditional methods, this research utilizes evidence-based prediction techniques, which have been shown to be more accurate and efficient. The use of these intelligent forecasting models enables local health authorities to create more precise and effective preventive measures, especially in the face of future outbreaks.

This study is particularly innovative in its use of hybrid forecasting models by machine learning for Malaysia’s future pandemics, such as avian flu or novel coronavirus strains. According to Moore [34], the scenario is for the next possible new pandemic of avian influenza virus strain H7N9 or a novel coronavirus. The predictive models developed are more precise, accurate, and efficient in anticipating the dynamic spread of the virus. This approach has been tested on real-world data, including daily new cases, daily new death cases, and daily new recovered cases of COVID-19, making it a valuable tool for public health officials and researchers. This research also has significant implications for future outbreaks, particularly in countries with tropical rainforests such as Malaysia. By predicting the spread of COVID-19 early on, this model can help policymakers build better healthcare facilities, take legislative action, and avoid economic losses. While a vaccine is now available, this model remains useful in accurately forecasting and preventing the impact of future pandemics, including those caused by new virus strains.

This study’s innovative and evidence-based methods make a valuable contribution to pandemic prediction and prevention, providing significant insights that can be used to mitigate the impact of future outbreaks. The implications of this research extend to public health authorities, policymakers, and researchers worldwide, offering powerful tools for mitigating the devastating effects of pandemics. The remainder of this paper is structured as follows. Materials and Methods goes into detail about the method we used to develop our proposed model. The hybrid ARIMA–SVM model used in this study is then briefly described. The results and discussion present the performance of our proposed model based on three known COVID-19 case datasets. Finally, we wrap up the article and make suggestions for future research.

## 2. Materials and Methods

### 2.1. ARIMA Modelling

The ARIMA (*p*, *d*, *q*) autoregressive integrated moving average model is one of the families in time-series forecasting that is widely used for time-series forecasting series datasets due to its flexibility with different time categories [16]. It also explicitly considers several standard patterns in time-series analysis, allowing for a powerful and easy-to-use way to produce accurate time-series forecasts. However, limitations may occur due to the existence of assumptions of a linear form that represents a linear relationship between the future value of the time series with the current value, the past value, and random noise in the model [15,16,17,21,26]. In the ARIMA model, p and q are the numbers of the autoregressive and moving average terms, and they are always listed in the order of the model, while d is the integer representing the differential order. The ARIMA model type with mean μ is represented mathematically as follows:(1)$$Y_{t}=\beta +\theta_{1}Y_{t-1}+\theta_{2}Y_{t-2}+\cdots +\theta_{p}Y_{t-p}+e_{t}-\emptyset_{1}e_{t-1}-\emptyset_{2}e_{t-2}-\cdots -\emptyset_{q}e_{t-q}$$
where $Y_{t}\;\mathrm{and}\;e_{t}$ are the actual value and the random error at time *t,* respectively. Both are assumed to be independently and identically distributed (*iid*) with a mean 0 and a constant variance of $\sigma^{2}$; $\theta_{i}\;\left(i=1,2,\ldots ,q\right)$ and $\emptyset_{j}\left(j=0,1,2,\ldots ,q\right)$ are the model parameters that need to be predicted.

### 2.2. Support Vector Machines Modelling

The Support Vector Machine (SVM) introduced by Vladimir Vapnik [18], which incorporates statistical learning theory, can handle larger dimensional data better, even with a small number of instances generalizability. Because the models select boundary support vectors from the input data, they process the data quickly. The SVM regression function is written as follows.

For linear and regressive dataset {$x_{i},Y_{i}$} the function is formulated as follows:(2)$$f\left(x\right)=W^{T}x+b$$

The coefficient W and *b* are estimated by minimizing.
(3)$$\frac{1}{2}W^{T}W+C\frac{1}{n}\overset{n}{\underset{i-1}{\sum }}L_{\varepsilon }\left(y_{i},f\left(x_{i}\right)\right)$$
where *ℓ*_ε_ is called the ε-intensive loss function and is formulated as follows:(4)$$\ell_{\varepsilon }\left(Y,f\left(x\right)\right)=\{\;\begin{array}{c} 0\;\mathrm{if}\;\left|Y-f\left(x\right)\right|\leq e\\ \left|Y-f\left(x\right)\right|\;\;\;\mathrm{others} \end{array}$$

Equation (3) can be transformed to the following constrained formulation by introducing positive slack variables ξ and $\xi_{i}^{*}$:(5)$$min\frac{1}{2}W^{T}W+C\overset{n}{\underset{i=1}{\sum }}\left(\xi_{i}+\xi_{i}^{*}\right)\;Wx_{i}+b_{i}-Y_{i}\leq \xi +\xi_{i}^{*}\;-Wx_{i}-b_{i}+Y_{i}\leq \xi +\xi_{i}^{*}\;\xi_{i},\xi_{i}^{*}\geq 0\;i=1,2,\ldots ,N$$

We always use dual theory to convert the above formula into a convex quadratic programming problem when solving it. Adding the Lagrange Equation (5) results in the following term:(6)$$min\frac{1}{2}\overset{n}{\underset{i,j=1}{\sum }}\left(\alpha_{i}^{*}-\alpha_{i}\right)\left(\alpha_{j}^{*}-\alpha_{j}\right)\alpha_{i}^{T}\alpha_{j}-\overset{n}{\underset{i=1}{\sum }}\alpha_{i}^{*}(Y_{i}-e)-\alpha_{i}\left(Y_{i}+e\right)$$

Subject to
$$\overset{Y}{\underset{i=1}{\sum }}\left(\alpha_{i}-\alpha_{i}^{*}\right)=0\;,\;\;\;\alpha_{i},\alpha_{i}^{*}\in \left[0,C\right]$$

When a dataset cannot be regressed linearly, we map it to a high dimension feature space and regress it linearly. The following is the formulation:(7)$$min\frac{1}{2}\overset{n}{\underset{i,j=1}{\sum }}\left(\alpha_{i}^{*}-\alpha_{i}\right)\left(\alpha_{j}^{*}-\alpha_{j}\right)\varphi \left(x_{i}{)}^{T}\varphi \left(x_{j}\right)-\overset{n}{\underset{i=1}{\sum }}\alpha_{i}^{*}(Y_{i}-e\right)-\alpha_{i}\left(Y_{i}+e\right)$$

Subject to
$$\overset{n}{\underset{i=1}{\sum }}\left(\alpha_{i}-\alpha_{i}^{*}\right)=0\;,\;\;\;\alpha_{i},\alpha_{i}^{*}\in \left[0,C\right]$$

Let $K\left(X_{i},X_{j}\right)=\left\{\varphi \left(X_{i}\right)\cdot \varphi \left(X_{j}\right)\right\}=\varphi^{T}\left(X_{j}\right)\varphi (X_{i});K\left(x,x\right)$ is the inner product of feature space and is called kernel function. Any symmetric function that satisfies Mercer condition can be used as Kernel function [19]. The Gaussian kernel function is specified in this study.
(8)$$K\left(x_{i},x_{j}\right)=exp(-||x_{i}-x_{i}|{|}^{2}/\left(2\sigma^{2}\right))$$

SVMs were used to estimate the nonlinear behaviour of the forecasting dataset because Gaussian kernels perform well under general smoothness assumptions [22].

### 2.3. Least-Square Support Vector Machines Modelling

The Least-Squares Support Vector Machines (LSSVM) proposed by Suykens and Vandewalle [35] is a modification of the standard SVM. LSSVM formulates the training process by solving linear problem quicker than SVM through quadratic programming. Additionally, this model is also more time efficient when analysing huge data. Consider a given training set $\left\{(x_{i},\;y_{j}),\;i=1,2,\ldots n\right\}$ with $x_{i}\in R^{n}$ as input data and $y_{i}\in R$ as output data. LSSVM defines the regression function as:(9)$$\min J\;\left(\omega ,e\right)=\frac{1}{2}\omega^{T}\omega +\frac{C}{2}\overset{n}{\underset{i=1}{\sum }}e_{i}^{2}$$

Subject to
$$Y_{i}=\omega^{T}\varphi (x_{i})+b+e_{i};\;\;i=1,2,3\ldots n$$
where ω is the weight vector; y is the regularization parameter where it determines the trade-off between the training error minimization and smoothness of the estimated function; e is the approximation error; φ(.) is the nonlinear function; and *b* is the bias term. Constrained optimization of Equation (9) can be translated to unconstrained optimization by constructing Lagrange function. This can be obtained by using Karush–Kuhn–Tucker (KKT) condition, where it partially differentiates with respect to ω,b,e, and φ(.):$$\left[\begin{array}{cc} 0 & I_{v}^{t}\\ I_{v} & \Omega +c^{-1}I \end{array}\right]\left[\begin{array}{c} b\\ a \end{array}\right]=\left[\begin{array}{c} 0\\ Y \end{array}\right]$$
where $Y={\left[Y_{1},\ldots ,Y_{n}\right]}^{T}$; $I_{v}={\left[1,\ldots ,1\right]}^{T}$; $a={\left[a_{1},\ldots ,a_{n}\right]}^{T}$; $\Omega =\left\{\Omega_{ij}|\;i,j=1\ldots n\right\}\;,\;\Omega_{ij}=\varphi {\left(x_{i}\right)}^{T}\varphi (x_{i})=K(x_{i},\;x_{j});\;K\left(.\right)$ is the Radial Basis Function (RBF) kernel function that obtains a and b by calculating linear operations.

### 2.4. Proposed Hybrid Model

Despite the various time-series models presented, the accuracy, efficiency, and precision of time-series forecasts are becoming crucial for many decision-making processes today. However, these factors do not appear in ARIMA and SVM models. This is also the main reason why the time-series forecasting model is crucial, more demanding, and dynamic, as well as actively researched in many fields of study. ARIMA and SVM models have also prevailed in their linear or nonlinear domains [15,25,26]. However, none of these are generic principles that can be generalized to all situations. Therefore, a hybrid approach using both linear and non-linear modelling capabilities is recommended. This approach is mainly proposed to improve the overall prediction effectiveness. Therefore, there is no research on how to improve the effectiveness of predictive models created in Malaysia, particularly in the case of COVID-19.

There are two reasons for using hybrid models in this study. First, a single ARIMA and SVM model may not be sufficient to identify all the time series’ characteristics. The second assumption is that one or both cannot recognise the actual data generation process. This study’s hybrid models were built in two stages. Part I discusses linear autocorrelation composition, and Part II discusses nonlinear components. Thus,
(10)$$Y_{t}=\ell_{t}+N_{t}$$
where $\ell_{t}$ and $N_{t}$ are denoted as the linear composition and the nonlinear component, respectively. Based on the data, these two parts must be approximated. Part I focuses on linear modelling, which employs the ARIMA model to model the linear composition. The model from the first model included residuals, which are nonlinear interactions that cannot be modelled by a linear model or possibly a linear relationship. Thus,
(11)$$\ell_{t}=\left[\overset{p}{\underset{i=1}{\sum }}\theta_{i}z_{t-i}-\overset{p}{\underset{i=1}{\sum }}\emptyset_{j}e_{t-j}\right]+\varepsilon_{t}={\hat{\ell }}_{t}+\varepsilon_{t}$$

Let $\varepsilon_{i}$ denoted as the residual from the linear model at time *t*. Then,
$$\varepsilon_{t}=Y_{t}-{\hat{\ell }}_{t}$$
where ${\hat{\ell }}_{t}$ is the predicted value for time *t* from the estimated relationship in (1), with $\varepsilon_{t}$ as the residual at time *t* from the linear model. The residual dataset after ARIMA fitting will only contain non-linear relationships that can be represented by a linear model [15]. The first stage results, which include forecast values and residuals from linear modelling, are then used in Part II.

Following Part II, the emphasis is on nonlinear modelling, where LSSVM is used to predict the nonlinear connection that occurs in residuals of linear modelling and original data. Then, the residual can be calculated using LSSVM by modelling various configurations as follows:

Part II focuses on nonlinear modelling, and LSSVM is used to model the nonlinear (possibly linear) relationship that occurs in residuals of linear modelling as well as original data. The residual can then be calculated using LSSVM by modelling different configurations as follows:(12)$$\varepsilon_{t}=f\;\left(\varepsilon_{t-1},\varepsilon_{t-2},\ldots \varepsilon_{t-n}\right)+e_{t}$$
(13)$$\varepsilon_{t}=f\;\left(\varepsilon_{t-1},\varepsilon_{t-12}\right)+e_{t}$$
(14)$$Y_{t}=f\;\left(Y_{t-1},Y_{t-12},\;\;{\hat{\ell }}_{t}\right)+e_{t}$$
(15)$$Y_{t}=f\;\left(Y_{t-1},Y_{t-12}\right)+e_{t}$$
where *f* is a nonlinear function determined by the LSSVMs model and $e_{t}$ is the random errors. Thus, the hybrid forecast is
(16)$${\hat{Y}}_{t}={\hat{\ell }}_{t}+\hat{N_{t}}$$

Equations (12) and (13) can be identified as $\hat{N_{t}}$, therefore the forecasted values can be achieved by summation of linear and nonlinear components. Figure 1 shows the functional flowchart of hybrid models.

In short, the proposed hybrid process methodology is divided into two parts. The ARIMA model is used to analyse the linear composition problem in Part I. Part II develops an LSSVM model to model the residuals from Part I. Because the ARIMA model in Part I cannot handle the nonlinear component of the data, the residuals of the linear model will include information about the nonlinearity. The LSSVM results can be utilised as forecasts of the ARIMA model’s error terms. The hybrid model defines various patterns by combining the distinct features and strengths of the ARIMA and LSSVM models. As a result, it is more effective to model linear and non-linear patterns separately with two different models and then re-hybridize the forecast results to improve overall modelling and forecasting performance.

### 2.5. Proposed Algorithm

**Step 1**: Three selected time series of COVID-19 cases datasets (1 October 2020–4 November 2022), namely daily new positive cases, daily new deaths cases, and daily new recovered cases, are generated in R programming Language.**Step 2**: Each of the generated datasets is defined as $\left\{X_{1i}=x_{11},x_{12},x_{13},\ldots ,\;x_{n1}\right\}$, $\left\{X_{2i}=x_{21},x_{22},x_{23},\ldots ,\;x_{2n}\right\},$ and $\left\{X_{3i}=x_{31},x_{32},x_{33},\ldots ,\;x_{3n}\right\}$ for daily new positive cases, daily new deaths cases, and daily new recovered cases, respectively. Then, the best ARIMA (*p*, *d*, *q*) is selected after checking the autocorrelation function (ACF) plot of ARIMA (*p*, *d*, *q*) residuals. The best fitted value for daily new positive cases is ARIMA (2, 1, 2), while it is ARIMA (1, 1, 2) and ARIMA (0, 1, 1) for daily new fatalities cases and daily new recovered cases of COVID-19, respectively.**Step 3**: The fitted value, $Y_{t-i}=(Y_{t-1}$, $Y_{t-2}$, …, $Y_{t-m})$ and the residuals $\varepsilon_{t-i}=\left(\varepsilon_{t-1},\;\varepsilon_{t-2},\;\ldots .\;,\;\varepsilon_{t-n}\right)$.**Step 4**: Combine the values in step 3 as a set of input variables to obtain the output $Y_{t}$**Step 5**: The ARIMA (*p*, *d*, *q*) is defined by the order of *q*. According to the information in step 4, Vector Machines is carried out to examine the residuals to obtain the output $L_{t}$ using R-programming Language.**Step 6**: A fitted value of ARIMA with the hybridization of Vector Machines model is obtained for all sample data. Then, the residuals $\varepsilon_{t}$ is generated to obtain the forecasting result $\hat{N_{t}}$.**Step 7**: The framing data split randomly into training data and testing data for further Vector Machines modelling. Run the Vector Machines procedure using the “e1071” and “liquidSVM” package in R-Programming Language.**Step 8**: The two modifiable parameters of the LSSVM technique (γ and σ) derived by objective function minimization such as mean square error (MSE). The grid-search method updates the parameters exponentially in the specified range using predetermined equidistant steps.**Step 9**: Assume the split data as the processing data and the order q as in Step 5. Therefore, the combine forecast as in Equation (16): ${\hat{Y}}_{t}={\hat{\ell }}_{t}+\hat{N_{t}}$**Step 10**: Estimate the model performance using the statistical measurement which are MSE, RMSE, MAE, and MAPE.

### 2.6. Forecasting Evaluation Criteria

In order to assess the overall performance of the proposed hybrid models, the one of a kind statistical measurements standard which accompanied by [15,16,36] including MAE (Mean Absolute Error), MAPE (Mean Absolute Percentage Error), MSE (Mean Squared Error), and RMSE (Root Mean Squared Error) are used.
$$\mathrm{MAE}=\frac{1}{n}\sum_{t=1}^{n}|\hat{Y_{t}}-Y_{t}|$$
$$\mathrm{MAPE}=\frac{1}{n}\sum_{t=1}^{n}\left|\frac{\hat{Y_{t}}-Y_{t}}{Y_{t}}\right|\times 100$$
$$\mathrm{MSE}=\frac{1}{n}\sum_{t=1}^{n}{\left(\hat{Y_{t}}-Y_{t}\right)}^{2}$$
$$\mathrm{RMSE}=\sqrt{\frac{1}{n}\sum_{t=1}^{n}{\left(\hat{Y_{t}}-Y_{t}\right)}^{2}}=\sqrt{\mathrm{MSE}}$$

In time-series analysis, measurement tools such as Akaike’s information criterion (AIC) and the Bayesian information criterion (BIC) are commonly used to determine the appropriate length for distributed lag for the ARIMA model. As a result, model selection is based on the model with the lowest AIC and BIC values to provide measures of model performance, resulting in the selection of the best ARIMA model. Meanwhile, three parameters such as *C* are used as measurement tools to determine the best fitted model for LSSVMs models. Meanwhile, for the LSSVMs models, two parameters such as γ and σ are used as the measurement tools to determine the best fitted model.

Incorrect LSSVM model parameter selection can lead to over or underfitting of the training data. The parameter sets of the LSSVMs model with the lowest MSE value, as with the ARIMA model, will be selected for use in the best fitting model. As a result, for the hybrid models, the ARIMA first functioned as a pre-processor, filtering the linear pattern of datasets. The ARIMA model’s error term is then fed into the SVM in the hybrid models. LSSVMs were used to reduce the ARIMA error function.

## 3. Results and Discussion

### 3.1. Application of the Hybrid Model of COVID-19 in Malaysia

This section examined the proposed model’s performance in two ways: first, the performance of the proposed models compared to ARIMA, SVM, LSSVM, ARIMA–SVM, and ARIMA–LSSVM models; second, the percentage improvement of the proposed models compared to ARIMA and SVM models. Since the World Health Organization (WHO) declared COVID-19 to be a worldwide pandemic, the COVID-19 time-series datasets have been extensively studied. The predictive capability of the developed novel models was then compared using three well-known datasets of daily COVID-19 cases in Malaysia—daily new positive cases data, daily new fatalities cases data, and daily new recovered cases data—to demonstrate the performance of the proposed model in terms of accuracy, effectively, and accurately. All these data are reported from the 1 October 2020 to 4 November 2022 and retrieved from the COVIDNOW website at https://covidnow.moh.gov.my/, accessed on 10 January 2023.

The minimum value of new death, new cases, and new recovered cases in Table 1 is 0, 2600, and 1.8, respectively, while the maximum value of new cases, death, and recovered cases is 33,872.0, 592, and 33,406, respectively. Similarly, the mean and median for new cases, deaths, and recovered cases are 6322.7, 47.51, and 6415.5, respectively, where the parentheses indicate the median (3471, 11, 3447.0). The first quartile values for daily new cases, death cases, and recover cases are 1922, 4, and 1843, respectively. The number of daily new cases, deaths, and recoveries in the third quartile is 6824, 58, and 6775, respectively. Furthermore, the standard deviations for new cases, deaths, and recoveries are 7097.8, 81.12, and 7058.3 percentiles, respectively.

Furthermore, this section discusses the process of proposed models for both parts, i.e., Part I (Linear Modelling) and Part II (Nonlinear Modelling), using three well-known COVID-19 datasets, namely daily new positive cases, daily new deaths cases, and daily new recovered cases, to demonstrate the effectiveness of the proposed models. Both linear and nonlinear modelling, as well as the data used in this study, are carried out using R programming.

Part I (Linear Modelling)—ARIMA is used to generate the best ARIMA model for the daily new positive case dataset (2, 1, 2). ARIMA is the best fitting ARIMA model for the daily new death case dataset (1, 1, 2). Meanwhile, the best ARIMA model is reported as ARIMA in the case of the daily new recovered cases dataset (0, 1, 1). Table 2 summarizes the results of this ARIMA (*p*, *d*, *q*) model. Table 3 displays the estimates for all parameters. The *p*-values for all parameters are small, as shown in this table. As a result, for confirmed, recovered, and death cases, the models were statistically significant and could be used to forecast the future [37,38].

Part II (Nonlinear Modelling)—Based on the concepts of support vector machine design and the use of pruning algorithms in R-programming software, an optimal machine learning algorithm was created. For the daily new positive COVID-19 cases datasets, parameters γ = 264, σ = 0.008 show the smallest values of MSE i.e., 6,661,412 (see Table 4). Therefore, this parameters value was selected for use in the best-fitting model for the datasets of daily new positive COVID-19 cases. Whereas the smallest value of MSE is 250.887 and 21114252 (Table 4), with parameters γ = 877, σ = 0.006 and γ = 334, σ = 0.008 are selected as the best-fitting model for daily new death and daily new recovered cases of COVID-19, respectively.

#### 3.1.1. New Positive Cases Data Forecasts

The daily new positive cases datasets series contains 765 data points and is recoded from 1 October 2020 to 4 November 2022 (see Figure 2). The number of daily new positive COVID-19 cases in Malaysia has increased significantly twice since July 2021, but has now dropped below 5000 new cases. However, it has continued and increased to a maximum of 33,406.00 around March–April 2022. This figure is expected to fall precipitously until 5 November 2022. The COVID-19 datasets have been extensively used with a wide range of linear and nonlinear time-series models, including ARIMA and machine learning methods [7,8,9,11,13,16,19,20,21,22,23,24,25,26]. The analysis of daily new positive cases of COVID-19 is critical as an indicator of the effectiveness of preventive measures that have been taken, are being taken, and will be taken by authorities to control the spread of this epidemic more effectively.

Therefore, a similar approach to that used by Aisyah et al. [15] is used to investigate the performance of the proposal models on daily new positive cases of COVID-19 datasets, where the dataset is divided into two samples, known as training sample and testing sample. According to Aisyah et al. [15] and Nurul Hila et al. [16], datasets should be divided into two parts to achieve the best results: 70–80% for training and the remaining 20–30% for testing [39,40]. The training data are used to assemble the models, while the testing data are used to evaluate the forecasting performances of the models based on statistical measurements. Thus, the daily new positive cases of the COVID-19 dataset are divided into two samples in this study: the training dataset and the test dataset. The training datasets contain 612 observations from day 1 to day 612, accounting for 80% of the datasets used exclusively to formulate from 1 October 2020 to 4 June 2022. In order to evaluate the forecasting performance of proposed models, the test sample datasets used approximately 153 observations from days 613–765 (20%) from the 5 June 2022 to the 4 November 2022

Table 5 displays the performance of the proposed model on the daily new positive COVID-19 case datasets. The proposed models produced results in terms of measurement error terms, namely MSE and MAE, which have smaller values of 10634.1142 and 46.54471, respectively. Similar results were obtained from the testing datasets, with MSE, MAPE, RMSE, and MAE values of 25478.114, 0.01547, 159.6182, and 75.6987, respectively. The findings are examined in greater detail using figures such as those shown in Figure 3a–e based on these numerical results. The estimated values for the proposed model (test sample) of daily new positive COVID-19 cases are shown in this figure. The proposed model line, as seen in this figure, closely matches the actual data. Figure 4a–e show the estimated values of our model for test sample data of ARIMA, SVM, LSSVM, ARIMA-SVM, and ARIMA–LSSVM models for COVID-19 cases. A comparison of the proposed model’s (ARIMA–LSSVM) lines for the test sample (Figure 4e) with the lines from the ARIMA, SVM, LSSVM, and ARIMA–SVM models (Figure 4a–d) clearly shows that the proposed model’s lines are somewhat close to the actual data. When we compared the performance of our proposal models to the performance of ARIMA, SVM, LSSVM, and ARIMA–SVM models, we discovered that our proposal models are efficient, accurate, and precise. In addition, the number of daily new positive COVID-19 cases is plotted, as shown in Figure 5. The daily new positive cases of COVID-19 for Malaysia are forecasted based on this figure for the next three weeks.

Based on Table 6, we examined the performance of the proposed models for the daily newly positive COVID-19 cases dataset by comparing the percentages of MSE, MAPE, RMSE, and MAE. The study hypothesis investigates the assumptions of the proposed hybrid model (ARIMA–LSSVM) approach to single ARIMA, SVM, LSSVM models, as well as hybrid ARIMA–LSSVM models. The proposed model outperformed the ARIMA–SVM model in MAE, MAPE, MSE, and RMSE, with improvements of 48.50%, 72.54%, 58.39%, and 35.50%, where the parentheses indicate an ARIMA, SVM, and LSSVM model, respectively, that results in the following: 80.96%, 89.80%, 91.48%, 70.81%; 80.61%, 89.97%, 90.72%, 69.54%; 63.20%, 79.59%, 69.31%, 44.60%. As a result of these findings (Table 5 and Table 6 and Figure 3, Figure 4 and Figure 5), it is possible to conclude that the proposed model produced greater accuracy and efficiency than ARIMA and SVM.

#### 3.1.2. New Deaths Cases Data Forecasts

In addition to the Malaysian daily new positive COVID-19 cases datasets, the Malaysian daily new deaths cases datasets are taken into account and used to evaluate the performance of the proposed models. This dataset, like the daily new positive dataset and the daily new death case dataset, has a recording period of 1 October 2020 to 4 November 2022 (see Figure 6) and contains 765 data points divided into two samples. As the number of daily positive COVID-19 cases reported rises, so does the number of deaths, which now stands at around 600. The training dataset contains 612 observations (80%) from 1 October 2020 to 4 June 2022, and the test sample contains approximately 153 observations (20%) from 5 June 2022 to 4 November 2022 to evaluate the prediction performance of the proposed model.

A similar approach to the daily new positive cases of the COVID-19 dataset was used to study the performance of the proposed model on the daily new death cases of the COVID-19 dataset. The dataset was divided into two samples, namely training sample and testing sample. It accounts for approximately 80% of the daily new death cases in the COVID-19 dataset for the training sample (involving 612 observations with the period 1 October 2020 until 4 June 2022). The remaining 20% is for the test sample, which includes approximately 153 observations from 5 June 2022 to 4 November 2022.

As shown in Table 7, the performance of the proposed models using the daily new deaths datasets from COVID-19 is first characterized by statistical measurements such as MSE, MAPE, RMSE, and MAE. The results for the training data in this table show that the proposed model produces the smallest MSE and MAE values of 19.6422 and 1.03218, respectively, when compared to ARIMA, SVM, LSSVM and ARIMA–SVM. The same pattern can be seen in the test data, where all the statistical measures used have the lowest values when compared to the ARIMA, SVM, LSSVM, and ARIMA–SVM models.

The study then examines the estimated value of the suggested model for the COVID-19 case dataset for daily deaths, as shown in Figure 7a–e. This graph makes it abundantly clear that the proposed model line and the observed data are nearly identical. Additionally, Figure 8a–e each show the estimated values for the test sample for ARIMA, SVM, LSSVM, ARIMA–SVM and ARIMA–LSSVM the suggested models. Once more, it is obvious that when compared to ARIMA, SVM, LSSVM, and ARIMA–SVM models, the test sample lines for our suggested model (Figure 8e) are somewhat close to the actual data. This demonstrates that the outcomes of our suggested model are in line with prior findings and are more effective, accurate, and precise than those of ARIMA, SVM, LSSVM, and ARIMA–SVM models. The number of daily COVID-19 death cases is also plotted, just like in Figure 9. The daily new death cases of COVID-19 in Malaysia are anticipated to decline because of this number over the course of the following three weeks, indicating a downward trend.

As shown in Table 8, a similar method in the daily addition of positive COVID-19 case dataset is used to investigate the performance of the proposed model for the daily recorded death COVID-19 case dataset using percentages MSE, MAPE, RMSE, and MAE. Again, the percentage of improvement shows that our proposed model outperforms the ARIMA–LSSVM models for all statistical measures, with results of 5.07%, 1.82%, 3.80%, and 1.98%, respectively; there are also improvements (62.47%, 67.03%, 85.31%, 61.68%; 61.61%, 66.74%, 83.92%, 59.90%; 61.01%, 65.10%, 83.50%, 59.34%) for MAE, MAPE, MSE, and RMSE. The ARIMA, SVM, LSSVM, and ARIMA–SVM model results are shown in parentheses. The presented results (see Table 7 and Table 8 and Figure 7, Figure 8 and Figure 9) clearly show that our proposed model outperforms the ARIMA, SVM, LSSVM, and ARIMA–SVM models in terms of efficiency and accuracy.

#### 3.1.3. New Recovered Cases Data Forecasts

The investigation to study the performance of the proposed model is continued with the dataset of new daily recovered cases of COVID-19 in Malaysia. Predicting Malaysia’s daily new recovered COVID-19 cases is just as important as the previous two datasets. The data used in this paper include daily observations from 1 October 2020 to 4 November 2022, for a total of 765 data points in the time series (Figure 10). The number of patients recovered from COVID-19 exhibits the same trend, with a significant increase twice. Beginning in July 2021, the number of recovered patients increases exponentially until it reaches over 22,500.00 in August 2021 (the time-series plot is shown in Figure 10) and then drops. 

However, around March–April 2022, the number of recovered COVID-19 cases increased again to a maximum of 33,872.00, then decreased and showed a relatively stable movement after that. This dataset is also divided into two samples, namely the training dataset and the test dataset. The training dataset, which included 612 observations (80%) from 1 October 2020 to 4 June 2022, was used in the same way as the previous datasets to formulate the model. In contrast, the test sample uses approximately 153 observations (20%) for the period 5 June 2022–4 November 2022. Table 9 displayed the performance of the proposed model on the daily new recovered COVID-19 case datasets based on training and testing samples. The results in Table 9 clearly show that the proposed training sample model produces the smallest MSE and MAE values, with 47602.551 and 80.2214, respectively, when compared to the MSE and MAE models of the ARIMA, SVM, LSSVM, and ARIMA–LSSVM models. For the test sample, the same scenario as the training sample produced the smallest MSE, MAPE, RMSE, and MAE with values of 13004.11, 0.0125, 114.0351, and 54.14471, respectively, when compared to ARIMA, SVM, LSSVM, ARIMA–SVM and ARIMA–LSSVM models.

Figure 11a–e shows the estimated value of the dataset for daily new recovered COVID-19 cases for the test sample. Once more, this graph demonstrates how closely the predicted value from the proposed models seems to match the actual values. Figure 12a–e present an additional analysis of the outcomes of the proposed model. These plots (Figure 12a–e) show the predicted values for the test samples derived from ARIMA, SVM, LSAVM, ARIMA–SVM and ARIMA–LSSVM models. In these models, however, the proposed model is close to the true value because, as we shall see in Figure 11e, the proposed model dominates them. As shown in Figure 13, the number of daily new recovered COVID-19 cases is plotted. This figure makes it abundantly clear that the suggested model maintains the data’s original sharpness. The daily new recovered COVID-19 cases for Malaysia are predicted from this figure for the upcoming three weeks, and it suggests that these cases will rise in the days to come in Malaysia.

As shown in Table 10, further research was completed to determine how well the proposed models performed for the daily newly recovered COVID-19 case datasets for MSE, MAPE, RMSE, and MAE in terms of the percentage. When comparing the results of the proposed model to ARIMA, SVM, LSSVM, and ARIMA–SVM models, the results show a better improvement when looking at the percentage of improvement for statistical measurements such as MSE, MAPE, RMSE, and MAE, with results of 47.99%, 68.43%, 50.20%, and 9.42% improvement (86.02%, 91.99%, 95.21%, 78.11%; 85.43%, 91.69%, 94.57%, 76.71%; 81.06%, 88.91%, 91.32%, 70.54%). The results reported in the parentheses are the ARIMA, SVM, LSSVM, and ARIMA–SVM models. As a result, based on the findings, the proposed model has produced results that are more accurate and effective than those produced by ARIMA, SVM, LSSVM, and ARIMA–SVM models.

## 4. Conclusions

In conclusion, predicting the spread of COVID-19 with accuracy and efficiency is essential but frequently challenging for decision-makers, especially the front-line workers and health care authorities. Despite what might seem to be an endless spread of COVID-19, there have been numerous efforts to develop time-series models and ongoing research to enhance forecasting model efficacy. One of the most well-liked types of hybrid models that divide time series into linear and non-linear forms is the hybrid approach. In this study, a hybrid model that combines some linear and non-linear predictions is proposed. Utilizing three well-known COVID-19 datasets—daily new positive cases, daily new death cases, and daily new recovered cases—revealed that our proposed models were demonstrated as having the highest efficiency, accuracy, and precision. In comparison to ARIMA, SVM, LSSVM, and ARIMA–SVM models, the proposed model with cross-validation check based on MSE, RMSE, MAE, and MAPE makes the most accurate predictions. In terms of performance (the proposed models compared to ARIMA, SVM, LSSVM and ARIMA–SVM models) for both the training and testing datasets, the proposed models’ performance yields the smallest values of MSE, RMSE, MAE, and MAPE. This indicates that the proposed model’s predicted value is more closely aligned with the observed value. Therefore, our proposed models had a higher level of precision and could be suggested for COVID-19 forecasting. It can be concluded that the proposed model may be the most efficient and effective way to increase prediction accuracy performance, especially since it is important to anticipate and stop the spread of COVID-19 cases.

## 5. Limitations and Future Recommendation

In this research study, an attempt was made to predict the overall number of confirmed cases, fatalities, and recoveries of COVID-19 in Malaysia. Investigating SVM performance with various kernel functions and developing the best hyperparameters for the SVM forecasting model can help to increase the forecast’s accuracy in upcoming work. Since only one-step-ahead forecasting is considered in this paper, multi-step forecasts can be centralised in subsequent work. It has been demonstrated that multi-step forecasts can greatly increase the trading system’s realism [41,42]. Additionally, to improve the performance of the model in terms of efficiency and accuracy of dataset prediction, hybrid approaches such as bootstrap and double bootstrap methods [16,43,44] can be considered in the hybridization of ARIMA and SVM. Given the dearth of researchers using bootstrap in daily COVID-19 forecasting cases, it is a reliable method. Numerous studies have demonstrated that the bootstrap resampling method yields a more precise estimate [45]. Future studies should also consider (i) the clinical and behavioural aspects such as actions, cognition, and emotions and (ii) the possibility of the underreporting of cases and deaths, as well as delays in notifications, in order to avoid biased predictions, forecasts, and results.

## Figures and Tables

**Figure 1 diagnostics-13-01121-f001:**
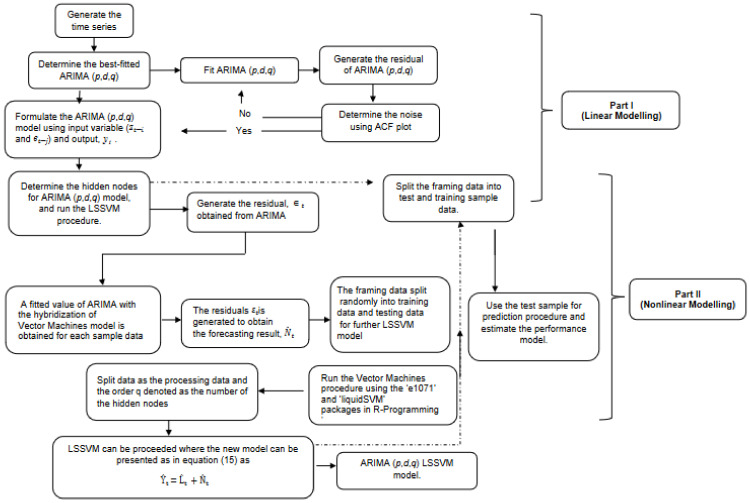
Flowchart process for hybrid ARIMA-LSSVM models.

**Figure 2 diagnostics-13-01121-f002:**
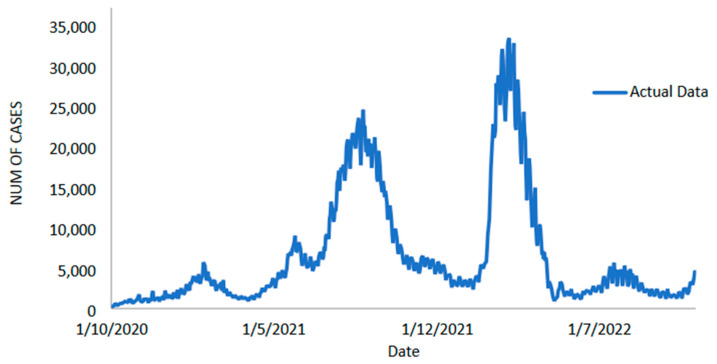
Malaysian daily new positive COVID-19 cases (1 October 2020 to 4 November 2022).

**Figure 3 diagnostics-13-01121-f003:**
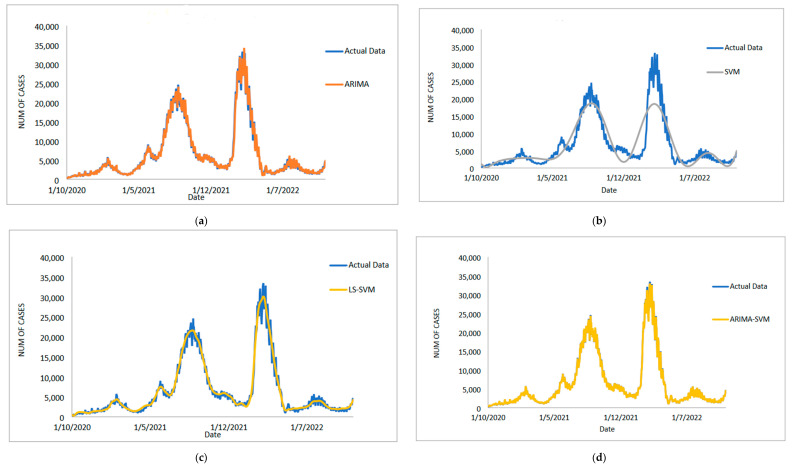
Results obtained from the proposed model for daily new positive COVID-19 cases dataset: (**a**) actual data vs. ARIMA model, (**b**) actual data vs. LSSVM models, (**c**) actual data vs. SVM model, (**d**) actual data vs. ARIMA–SVM models, (**e**) actual data vs. ARIMA–LSSVM models.

**Figure 4 diagnostics-13-01121-f004:**
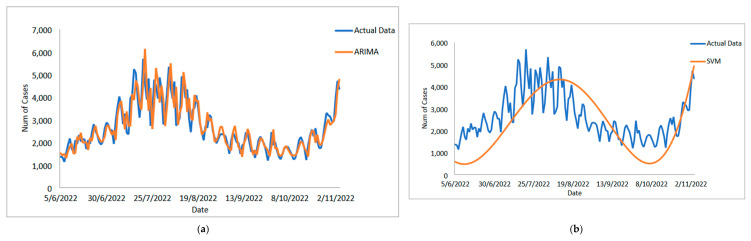
Models’ prediction of daily new positive COVID-19 cases dataset (20% test sample): (**a**) actual data vs. ARIMA model, (**b**) actual data vs. SVM model, (**c**) actual data vs. LSSVM models, (**d**) actual data vs. ARIMA–SVM models, (**e**) actual data vs. ARIMA–LSSVM models.

**Figure 5 diagnostics-13-01121-f005:**
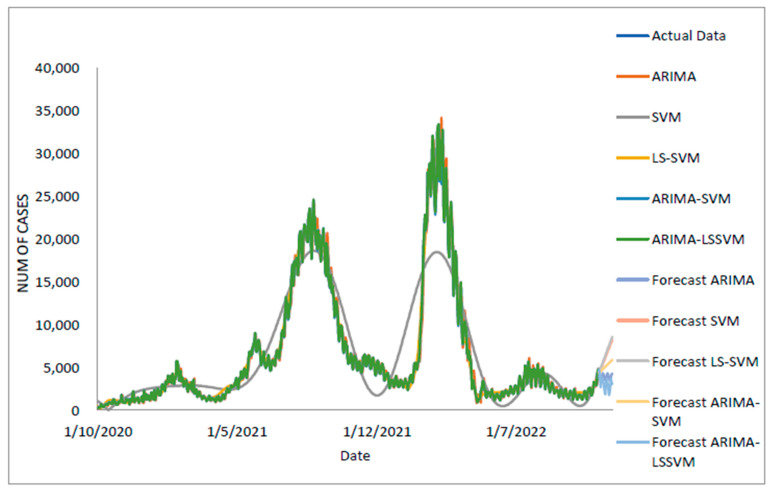
Actual and three weeks ahead forecasted values of ARIMA, SVM, LSSVM, ARIMA–SVM, and ARIMA–LSSVM models for new cases of COVID-19 of the 80% training and 20% testing set.

**Figure 6 diagnostics-13-01121-f006:**
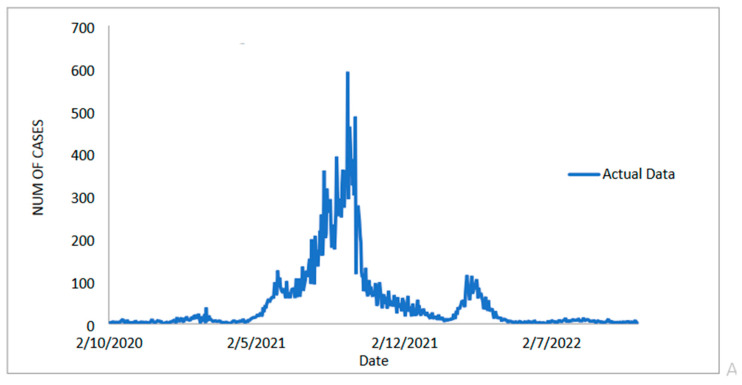
Malaysian daily new deaths COVID-19 cases (1 October 2020 to 4 November 2022).

**Figure 7 diagnostics-13-01121-f007:**
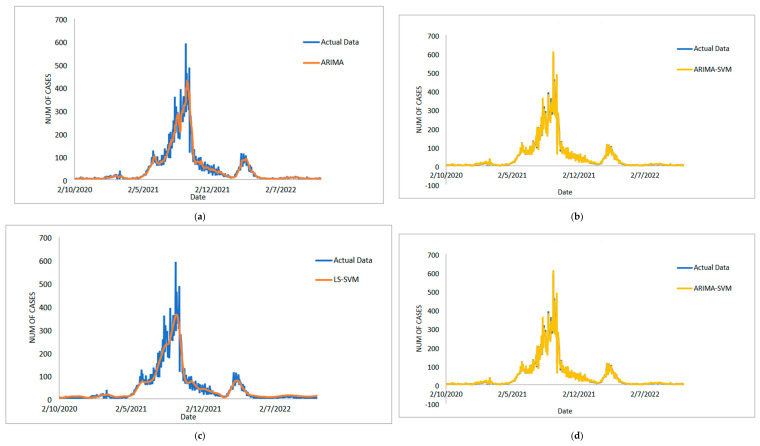
Results obtained from the proposed model for daily new death COVID-19 cases dataset: (**a**) actual data vs. ARIMA model, (**b**) actual data vs. LSSVM models, (**c**) actual data vs. SVM model, (**d**) actual data vs. ARIMA–SVM models, (**e**) actual data vs. ARIMA–LSSVM models.

**Figure 8 diagnostics-13-01121-f008:**
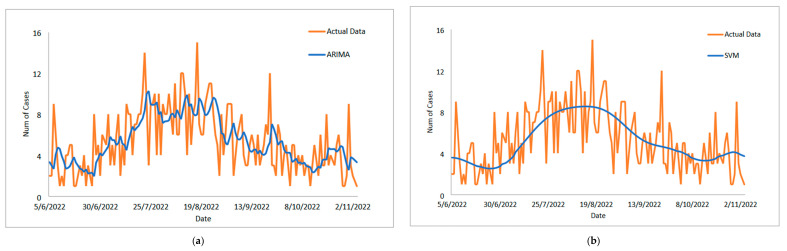
Models’ prediction of daily new death COVID-19 cases dataset (20% test sample): (**a**) actual data vs. ARIMA model, (**b**) actual data vs. SVM model, (**c**) actual data vs. LSSVM models, (**d**) actual data vs. ARIMA–SVM models, (**e**) actual data vs. ARIMA–LSSVM models.

**Figure 9 diagnostics-13-01121-f009:**
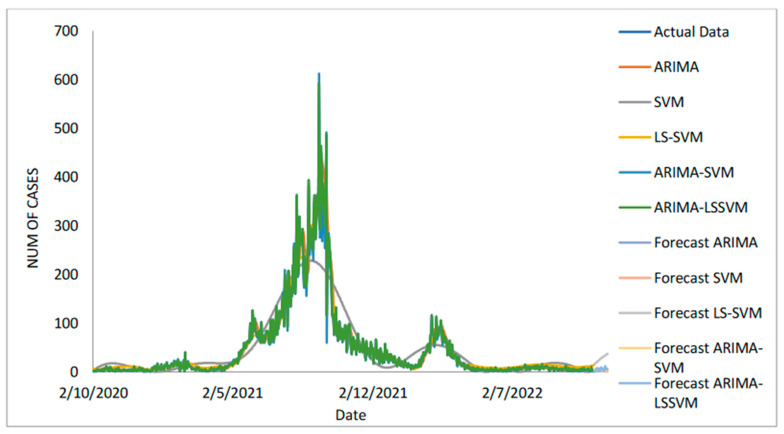
Actual and three-weeks-ahead forecasted values of ARIMA, SVM, LSSVM, ARIMA–SVM and ARIMA–LSSVM models for daily new deaths COVID-19 cases of the 80% training and 20% testing set.

**Figure 10 diagnostics-13-01121-f010:**
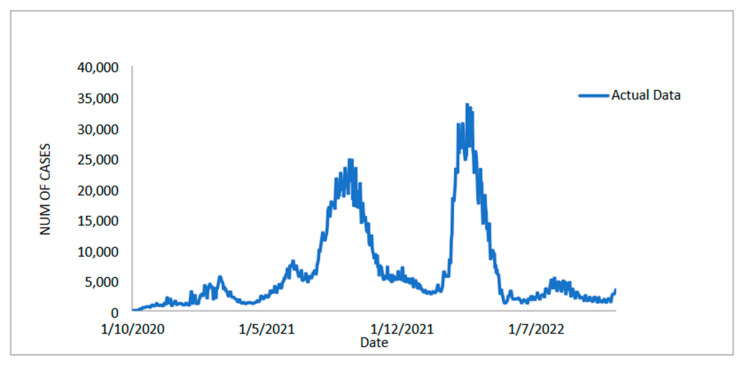
Malaysian daily new recovered COVID-19 cases (1 October 2020 to 4 November 2022).

**Figure 11 diagnostics-13-01121-f011:**
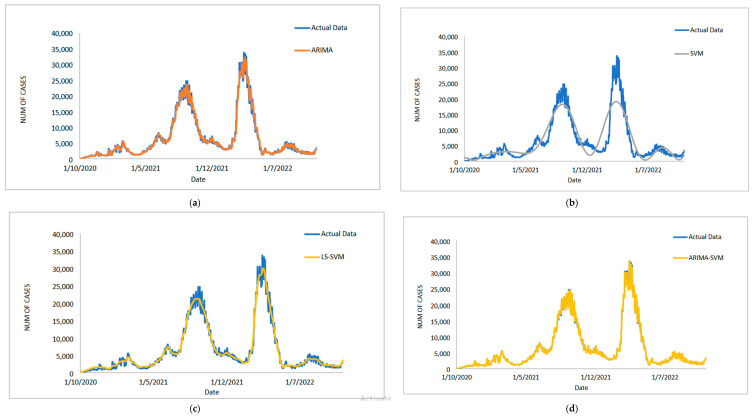
Results obtained from the proposed model for daily new recovered COVID-19 cases dataset: (**a**) actual data vs. ARIMA model, (**b**) actual data vs. LSSVM models, (**c**) actual data vs. SVM model, (**d**) actual data vs. ARIMA–SVM models, (**e**) actual data vs. ARIMA–LSSVM models.

**Figure 12 diagnostics-13-01121-f012:**
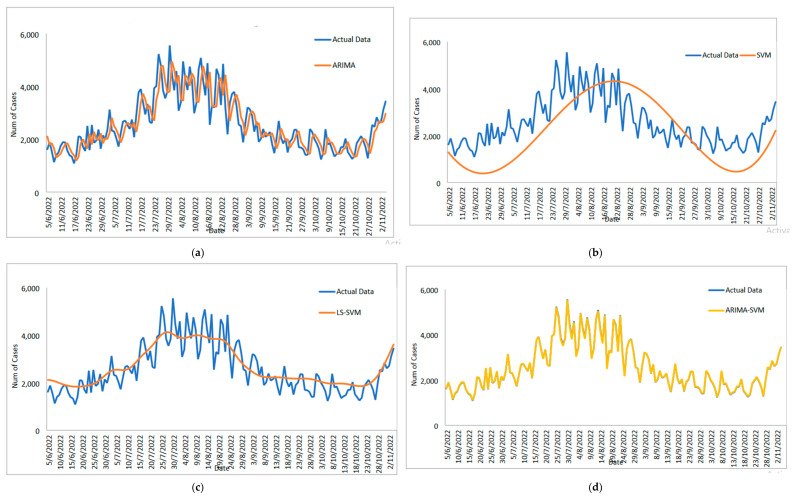
Models’ prediction of daily new recovered COVID-19 cases dataset (20% test sample): (**a**) actual data vs. ARIMA model, (**b**) actual data vs. SVM model, (**c**) actual data vs. LSSVM models, (**d**) actual data vs. ARIMA–SVM models, (**e**) actual data vs. ARIMA–LSSVM models.

**Figure 13 diagnostics-13-01121-f013:**
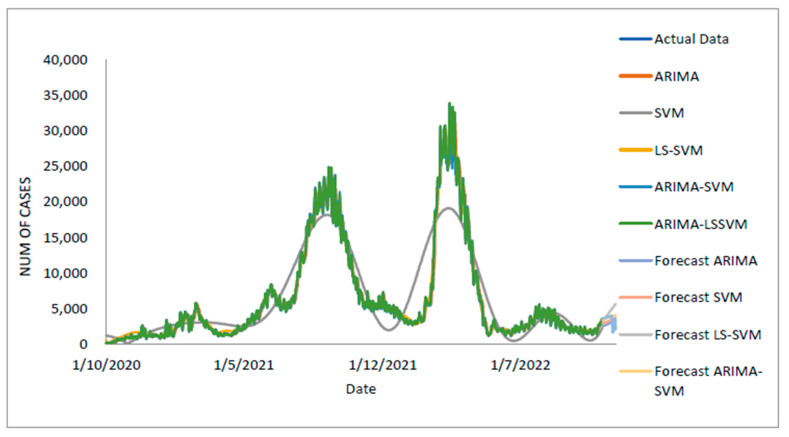
Actual and three weeks ahead forecasted values of ARIMA, SVM, LS–SVM, ARIMA–SVM and ARIMA–LSSVM models for daily new recovered COVID-19 cases of the 80% training and 20% testing set.

**Table 1 diagnostics-13-01121-t001:** Descriptive statistics of COVID-19 daily new cases, death, and recovered cases of Malaysia.

	New Case	New Death	New Recovered
Min	2.60000 × 10^2^	0	1.8
1st Qu	1.9220 × 10^3^	4	1.8430 × 10^3^
Median	3.4710 × 10^3^	11	3.4470 × 10^3^
Mean	6.4155 × 10^3^	47.5098	6.3227 × 10^3^
3rd Qu	6.8240 × 10^3^	58	6.7750 × 10^3^
Max	3.3406 × 10^4^	592	3.3872 × 10^4^
SD	7.0978 × 10^3^	81.1215	7.0583 × 10^3^

**Table 2 diagnostics-13-01121-t002:** The best ARIMA (*p, d, q*) model selection.

COVID-19 Daily Cases	ARIMA (*p*, *d*, *q*)	AIC	BIC
Daily New Positive Cases	(2, 1, 2)	12,564.54	12,587.73
Daily New Deaths Cases	(1, 1, 2)	6930.12	6948.63
Daily New Recovered Cases	(0, 1, 1)	13,044.74	13,054.01

**Table 3 diagnostics-13-01121-t003:** Parameter estimates of ARIMA models and their *p*-values.

Model Parameters	Estimate	Z-Stat	*p*-Value
New Case ARIMA (2, 1, 2)			
$\theta_{1}$	1.2408	120.085	2.2 × 10^−16^
$\theta_{2}$	−0.9715	−98.320	2.2 × 10^−16^
$\varphi_{1}$	−1.2628	−42.225	2.2 × 10^−16^
$\varphi_{2}$	0.8738	48.102	2.2 × 10^−16^
Recovered Case ARIMA (0, 1, 1)			2.2 × 10^−16^
$\varphi_{1}$	−0.3473	−9.953	2.2 × 10^−16^
Death Case ARIMA (1, 1, 2)			2.2 × 10^−16^
$\theta_{1}$	0.8595	19.852	2.2 × 10^−16^
$\varphi_{1}$	−1.6196	−35.651	2.2 × 10^−16^
$\varphi_{2}$	0.7039	20.432	2.2 × 10^−16^

**Table 4 diagnostics-13-01121-t004:** LSSVMs Model Parameters for the daily new COVID-19 cases datasets.

COVID-19 Daily Cases	LSSVM Parameter	MSE
	γ = 11, σ = 0.008	11,432,512
	γ = 38, σ = 0.008	10,235,488
Daily New Positive Cases	γ = 74, σ = 0.008	9,025,413
	γ = 110, σ = 0.008	8,014,123
	γ = 264, σ = 0.008	6,661,412
	γ = 25, σ = 0.006	1678.364
	γ = 56, σ = 0.006	1233.481
Daily New Deaths Cases	γ = 277, σ = 0.006	965.143
	γ = 436, σ = 0.006	554.368
	γ = 877, σ = 0.006	250.887
	γ = 54, σ = 0.008	28,412,113
	γ = 89, σ = 0.008	27,140,039
Daily New Recovered Cases	γ = 125, σ = 0.008	26,412,142
	γ = 275, σ = 0.008	23,032,256
	γ = 334, σ = 0.008	21,114,252

**Table 5 diagnostics-13-01121-t005:** Performance measures of the proposed model for daily new positive COVID-19 cases datasets.

MODELS	TRAIN		TEST			
	MSE	MAE	MSE	MAPE	RMSE	MAE
ARIMA	929,843.169	611.0274	298,988.28	0.15167	546.7982	397.57
SVM	8,355,184.483	2001.644	274,588.16	0.15421	524.0116	390.3848
LSSVM	1084.1527	739.5387	83,026.550	0.07580	288.1432	205.6450
ARIMA–SVM	42,552.7137	90.34845	61,223.474	0.05633	247.4337	146.9841
ARIMA–LSSVM	10,634.1142	46.54471	25,478.114	0.01547	159.6182	75.6987

**Table 6 diagnostics-13-01121-t006:** Percentage improvement of the proposed models with other forecasting models (the COVID-19 cases of daily new positive cases).

Model	MAE	MAPE	MSE	RMSE
ARIMA	80.9596549	89.80022417	91.47855762	70.80857252
SVM	80.60920917	89.96822515	90.72133554	69.53918577
LSSVM	63.18962289	79.59102902	69.31329316	44.60455773
ARIMA–SVM	48.49871517	72.5368365	58.38505669	35.49051726

**Table 7 diagnostics-13-01121-t007:** Performance measures of the proposed model for daily new deaths COVID-19 cases datasets.

MODELS	TRAIN		TEST			
	MSE	MAE	MSE	MAPE	RMSE	MAE
ARIMA	697.999	11.8083	6.06741	0.56838	2.46321	1.92791
SVM	1409.19	21.8006	5.38920	0.53687	2.32146	1.85605
LSSVM	505.181	11.4309	5.38920	0.53687	2.32146	1.85605
ARIMA–SVM	49.4459	3.53812	0.92630	0.19088	0.96303	0.76230
ARIMA–LSSVM	19.6422	1.03218	0.89114	0.18741	0.94400	0.72364

**Table 8 diagnostics-13-01121-t008:** Percentage improvement of the proposed models with other forecasting models (the COVID-19 cases of daily new death cases).

Model	MAE	MAPE	MSE	RMSE
ARIMA	62.46505283	67.02734086	85.31267872	61.67602437
SVM	61.60592539	66.73588924	83.92334934	59.90434808
LSSVM	61.01182619	65.09210796	83.46433608	59.33593514
ARIMA–LSSVM	5.071494162	1.81789606	3.795746518	1.976054744

**Table 9 diagnostics-13-01121-t009:** Performance measures of the proposed model for daily new recovered COVID-19 cases datasets.

MODELS	TRAIN		TEST			
	MSE	MAE	MSE	MAPE	RMSE	MAE
ARIMA	1,802,678.36	804.4378	271,462.22	0.1560	521.0203	387.2768
SVM	7,636,804.13	1890.917	239,672.00	0.1504	489.5630	371.6573
LSSVM	1,206,113.52	723.9413	149,871.53	0.1127	387.1324	285.9190
ARIMA–SVM	99,205.699	136.8519	26,108.02	0.0396	161.5797	104.1002
ARIMA–LSSVM	47,602.551	80.2214	13,004.11	0.0125	114.0351	54.14471

**Table 10 diagnostics-13-01121-t010:** Percentage improvement of the proposed models with other forecasting models (the COVID-19 cases of daily new recovered cases).

Model	MAE	MAPE	MSE	RMSE
ARIMA	86.01911863	91.98717949	95.20960596	78.11311767
SVM	85.43154944	91.68882979	94.57420558	76.70675684
LSSVM	81.06291992	88.90860692	91.32316191	70.54364347
ARIMALSSVM	47.98789051	68.43434343	50.19112901	29.42485968

## Data Availability

Not applicable.

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
