# Peer review of "Improvement of Time Forecasting Models Using Machine Learning for Future Pandemic Applications Based on COVID-19 Data 2020–2022"

_diagnostics, 2023, doi:10.3390/diagnostics13061121_

Round 1
Reviewer 1 Report
1. If possible, avoid acronyms in the abstract.
2. State clearly the list of contributions in the introduction.
3. Several relevant paper on this topic should be added to the background, especially the papers that propose hybrid machine learning - swarm intelligence solutions. Please include the following:
https://www.mdpi.com/2075-1680/11/8/410
https://www.sciencedirect.com/science/article/pii/S2210670720308842
https://www.sciencedirect.com/science/article/pii/S095741742201764X
4. Concerning that there are numerous publications about this exact topic, utilizing a variety of machine learning models and solutions to predict the number of COVID-19 cases, elaborate and justify in details the novelty of the approach and how it contributes to the field.
5. Make sure that all the parameters in all equations have been explained in the text.
6. Figures should be provided in higher quality (i.e. Fig. 1 is not clear enough, same applies for Fig. 2 and Fig. 6).
7. Please use scientific annotation (10^x) instead of exp (e^x), this is I believe a requirement from the MDPI.
8. In the figures with graphs, the legend is oddly placed, with lines going over the graphs. Was that intended - if not, please remove these lines.
9. Consider testing the model on the additional dataset, to validate the performance and establish confidence in the method. There are numerous available open datasets, for example, for influenza cases. You could then generalize the model to other diseases as well.
10. Summarize the experimental results briefly in the conclusion.
Author Response
Thank you very much for your evaluation and giving us an opportunity to ‘revise and resubmit’ the manuscript titled as, “Improvement of time forecasting models using machine learning for
future pandemic applications based on COVID-19 data 2020-2022” with manuscript ID: diagnostics-2205707.

Author Response

(The authors gave the same response as above.)

Round 2
Reviewer 1 Report
Authors have addressed all my concerns from the previous round, and the paper can be accepted in its present form.